# Water, sanitation, and hygiene (WASH) factors and the incidence of communicable diseases in Urban Bangladesh: Evidence from municipal areas

Mohammad Hayatun Nabi[1], Fatema Hashem Rupa[ID][2]*, Ishrat Jahan[3], A. B. M. Nahid Hasan[4], Farah Naz[5], Iqbal Masud[6], Mohammad Delwer Hossain Hawlader[1], Mosharop Hossian[ID][7]

1 Department of Public Health, North South University, Dhaka, Bangladesh, 2 Directorate General of Health Services (DGHS), Dhaka, Bangladesh, 3 Faculty of Medicine, University of Queensland, Brisbane, Australia, 4 Department of Public Health Nutrition, Primeasia University, Dhaka, Bangladesh, 5 Department of Physiology, Green Life Medical College, Dhaka, Bangladesh, 6 Dhaka Ahsania Mission, Dhaka, Bangladesh, 7 Public Health Promotion and Development Society (PPDS), Dhaka, Bangladesh

* fatemarupa125@gmail.com

## Abstract

### Background

Safe drinking water, sanitation, and hygiene (WASH) are crucial for health and development, especially in low- and middle-income countries (LMICs) where WASH-related diseases are prevalent. Bangladesh, with high poverty rates and a large population, faces significant WASH challenges. This study examines the association between hygiene-related factors and communicable diseases in two Bangladeshi municipalities.

### Method

A descriptive cross-sectional study was conducted from June 1 to July 30, 2021, in Savar and Satkhira municipalities, chosen for their diverse socioeconomic and geographic characteristics. A stratified random sampling method targeted poor, marginalised, and socially excluded populations. A total of 607 interviews were conducted using a structured questionnaire, collecting data on socio-demographics, hygiene practices, and disease history. Participants were included if they were adults (aged 18 or older), had resided in the municipality for at least one year, and provided informed consent to participate. Data were analysed using SPSS, employing chi-square tests and logistic regression to assess associations between variables.

### Results

The study revealed that 24.1% of participants had a communicable disease, with diarrhoea being the most common. Significant predictors of communicable diseases

**Data availability statement:** All relevant data are within the paper and its Supporting Information files.

**Funding:** The author(s) received no specific funding for this work.

**Competing interests:** The authors have declared that no competing interests exist.

included gender, with females being more likely affected (OR: 3.21, 95% CI: 1.19 to 8.66, p = 0.003), and the source of drinking water, with tube well users at higher risk (OR: 2.81, 95% CI: 1.13 to 7.02, p = 0.015). Poor hand hygiene significantly increased disease risk (OR: 12.31, 95% CI: 2.86 to 53.03, p < 0.001).

## Conclusions

Communicable diseases are prevalent in Bangladeshi municipalities, primarily driven by poor WASH conditions. Enhancing access to safe drinking water and implementing community-based WASH interventions, such as hygiene education campaigns, improved sanitation infrastructure, and school-based awareness programs, are vital for reducing the disease burden. Further research should explore the effectiveness and scalability of these targeted strategies.

### Author summary

Access to safe water, adequate sanitation, and good hygiene (WASH) is essential for preventing communicable diseases, particularly in low- and middle-income countries like Bangladesh. Despite progress, many communities in Bangladesh still lack these basic services, leading to frequent outbreaks of diarrhoeal diseases and other infections. In this study, we investigated the link between hygiene-related factors and the occurrence of communicable diseases in two urban municipalities—Savar and Satkhira—using data from 607 residents. Our findings revealed that nearly one in four participants reported a communicable disease, with diarrhoea being the most common. Women and those using tube wells for drinking water were at significantly higher risk. Poor hand hygiene practices showed the strongest association with disease. These results highlight the urgent need for targeted WASH interventions, such as hygiene education, improved access to clean water, and better sanitation infrastructure. Promoting handwashing and safe water practices can substantially reduce disease risk. Our findings provide evidence to guide public health policy and community programs aimed at reducing WASH-related disease burdens in similar urban settings.

## Introduction

Water, sanitation, and hygiene (WASH) are critical for health and development [1]. They are considered a significant global public health issue [1]. Approximately 2.3 billion people still lack basic sanitation, and over half a billion lack access to adequate water sources globally [2]. In the 21st century, WASH remains a critical public health need in low- and middle-income countries (LMICs) [3]. Approximately half of the population in these countries suffers from diseases or infections caused by contaminated water and improper sanitation [4]. Infectious diseases are among the most prevalent causes of mortality globally [5]. They are the primary contributors to

disability-adjusted life years (DALYs) [6]. Key infectious diseases responsible for global fatalities include acute lower respiratory tract infections, HIV/AIDS, diarrhoeal diseases, tuberculosis, and malaria [6].

The WASH issues are particularly severe in certain countries, and Bangladesh is one such country facing considerable challenges. With the eighth largest population globally [7], Bangladesh also has the highest percentage of people living below the national poverty line among all South Asian countries [8]. Despite its significant population size, Bangladesh struggles with considerable challenges in meeting basic healthcare needs [9]. Research indicates a direct link between WASH score and communicable diseases, such as diarrhoea, typhus, pneumonia, and skin infections [10–12]. These conditions pose a threat to public health in Bangladesh and significantly contribute to the global disease burden.

Improved hygiene practices are considered among the most cost-effective social interventions for reducing the burden of diarrhoeal and other communicable diseases. Hand washing is the most prominent and affordable hygiene practice among low-income populations [13]. Hand hygiene is an easy way to prevent the spread of infections. The WHO started a global hand hygiene campaign in 2005 called "Clean Care is Safer Care" [14]. Bangladesh actively engaged with this initiative and participated in a WHO regional workshop held in Bangkok, Thailand, in 2007, alongside several other countries. Since then, hand hygiene principles have been progressively integrated into the country's infection prevention and control strategies, particularly within hospitals. The campaign's significance was further reinforced during the COVID-19 pandemic through nationwide awareness and behavioural change programs.

However, the WASH situation in Bangladesh remains a significant challenge, despite progress in some areas. While 99.5% of households have access to basic water, only 60.7% have access to basic sanitation, and 56.3% have access to basic hygiene [15]. A substantial portion of the population lacks access to safe drinking water and sanitation, and many do not practice basic hygiene, leading to health concerns such as diarrhoeal diseases. The consequences of poor WASH scores are reflected in the high incidence of various infectious diseases in Bangladesh. Bangladesh has a notable incidence of diarrhoeal diseases, tuberculosis, dengue fever, and various other infectious ailments [16]. While the prevalence of tuberculosis (TB) has declined substantially, Bangladesh still ranks among the top ten countries in the world with the highest TB burden [17]. Regarding the human immunodeficiency virus (HIV), only less than 0.01% of the population is reported to be HIV-positive [18], but rates are much higher among high-risk populations such as injecting drug users, sex workers, and men who have sex with men [19]. Pneumonia and waterborne diseases are widely prevalent [20,21]. Diarrhea, which spreads quickly due to its short incubation period, is the most prevalent disease. It is estimated that approximately 88% of diarrhoeal diseases are attributable to drinking contaminated water, improper sanitation, and poor hygiene [22].

Improving WASH practices can significantly reduce disease burden. An earlier estimation suggested that nearly 2.4 million deaths (4.2% of all causes of death) worldwide could be avoided by adopting appropriate WASH practices [23,24]. Despite being a densely populated LMIC, Bangladesh has achieved substantial improvements in basic WASH services, such as improved drinking water, hand washing, and the elimination of open defecation. However, a significant proportion of the population is still exposed to unsafe WASH conditions [25,26]. Different studies have shown that hand washing can decontaminate hands and prevent cross-transmission [27,28]. Washing hands with soap can also reduce the risk of endemic diarrhea, respiratory diseases, and skin infections [29]. Although previous studies have examined the effects of large-scale WASH interventions in rural Bangladesh [30] and explored gaps between hygiene knowledge and practice [31], most existing research remains rural-focused or intervention-based.

To address this gap, the present study investigates the association between hygiene-related factors and communicable diseases in two urban municipalities, Savar and Satkhira, selected for their contrasting socioeconomic and geographic characteristics. Community-level epidemiological studies in urban Bangladesh remain limited, yet understanding the relationship between hygiene behaviour and disease burden is essential. This study aims to generate evidence that can inform the design of targeted public health strategies and guide future policy development.

## Methodology

### Ethics statement

Ethical approval (2021/OR-NSU/IRB-No.0801) for the study was obtained from the Institutional Review Board of North South University Bangladesh. Participants were informed of the study's purpose, procedures, potential risks, and benefits. Written consent was obtained from all participants to ensure voluntary participation. Confidentiality of participant information was maintained throughout the study.

### Study design and setting

A descriptive cross-sectional study was conducted between 1st June and 30th July 2021 in two municipality areas of Bangladesh: Savar from Dhaka Division and Satkhira from Khulna Division. These areas were chosen due to their contrasting socioeconomic, geographic, and infrastructural profiles, which provide a broader representation of urban Bangladesh. Savar Municipality, located in the Dhaka District, is a rapidly urbanising area with industrial and peri-urban characteristics, representing high-density urban expansion. In contrast, Satkhira Municipality, situated in the southwestern part of the country, reflects a semi-urban context with relatively limited infrastructure and public health services. Together, these sites capture diverse urban settings and are therefore suitable for understanding variations in hygiene practices and disease burden across urban populations. Savar has a population of approximately 384,093 within an area of 14.08 km². Satkhira Municipality, located in the Khulna Division, has a population of approximately 113,322 within an area of 32.39 km².

### Sample selection

The targeted population included poor, extremely poor, marginalised, and socially excluded people from these urban communities.Low-income neighbourhoods were identified through local records and community health worker input.

- The income categorisation was established considering the urban municipal setting of the study and the average household size in Bangladesh, which is 4.18 members per family in urban areas [32]. In such contexts, a monthly household income of ≤ BDT 10,000 offers limited capacity to meet basic needs such as food, rent, healthcare, and education, and thus was classified as extremely poor. Households earning between BDT 10,001–20,000/month were categorised as poor, as they often remain economically vulnerable despite being above the lower poverty threshold. These thresholds reflect the practical cost of living in urban municipalities and align with national social safety net program criteria and poverty line estimations used by government and development agencies.

- **Marginalised** included informal workers, female-headed households, ethnic minorities, and people with disabilities.

A stratified random sampling method was used to ensure representation across the different sociodemographic groups. Interviewers approached the households systematically and provided a brief explanation of the study's objective, purpose, and eligibility criteria.

**Inclusion criteria**: Participants were included in the study if they met the following conditions:

- **Aged 18 years or older**: to ensure participants could provide informed consent and relevant health history.

- **Resident of the municipality for at least one year**: to ensure familiarity with local hygiene practices and reduce recall bias;

- **Willing to participate and provide informed consent**: to comply with ethical standards and ensure voluntary participation.

Individuals who declined to participate or did not meet the inclusion criteria were excluded. Within each household, the preferred respondent was the head of household or an adult member (≥18 years) who was knowledgeable about

the family's hygiene practices and health history. When the head was unavailable, the next most informed adult was interviewed. Ultimately, a total of 607 interviews were conducted. This sample size was chosen to achieve 80% power to detect a moderate effect size (Cohen's w = 0.3), at a significance level of 0.05. Using standard sample size formulas for chi-square tests, the initial calculation suggested a base sample size of 164 participants. To account for the complexity of the analyses, including multivariable regression models, the sample size was adjusted for a conservative adjustment factor approximately four times the initial estimate to ensure robust and reliable results.

## Data collection

**Data collection tool.** A structured questionnaire was developed based on the World Health Organization (WHO) Core Questions on Water, Sanitation and Hygiene for Household Surveys framework, which is widely used for assessing hygiene-related practices in LMICs. The tool was validated through expert reviews for content and cultural relevance. The questionnaire was initially written in English and translated into Bengali by a language expert. The translation process included back translation and pre-testing with a sample of native speakers to ensure accuracy and cultural relevance.

**Pilot testing.** Pilot testing was conducted with 5% of the study sample (approximately 30 participants), leading to minor adjustments in the wording of questions for clarity. For instance, the term *"excreta disposal"* was replaced with *"how your family disposes of toilet waste"*, and *"primary water source"* was rephrased as *"main source of drinking water for your household"*. These changes were made to ensure better understanding among participants with varying educational backgrounds.

**Data collection procedure.** Data were collected via face-to-face interviews conducted by trained interviewers. The participants were informed of the study's objective, purpose, and eligibility criteria.

## Measures

Participants provided information on their:

- **Socio-demographic profile:** age, gender, religion, highest level of education, employment, marital status.

- **Household characteristics:** monthly family income and household type, which referred to the structural condition of the dwelling (e.g., raw/mud, tin shed, semi-building, or full concrete building).

- **WASH practices:** assessed both access (e.g., source of drinking water, type of toilet) and behaviours (e.g., hand washing practices, waste disposal methods).WASH indicators in this study were interpreted and reported in line with the WHO/UNICEF Joint Monitoring Programme (JMP) service ladders for SDG 6. For water, "tube well" was considered a basic service under JMP, while "supply water" refers to piped water to premises, falling under safely managed services if quality and availability were adequate. For sanitation, "sanitary latrine" refers to a toilet that hygienically separates human excreta from contact (JMP definition of improved sanitation), while "ring slab with water seal" typically aligns with limited or basic sanitation depending on sharing status and quality. Handwashing practices were assessed in terms of presence and use of soap and water, as per JMP definitions of basic hygiene services.

- **Disease profile:** participants were asked to report their history of communicable and chronic diseases over the past 12 months. While a one-year recall period may introduce some degree of recall bias, it was chosen to capture a sufficient number of disease events, particularly for conditions like diarrhoea, typhoid, and jaundice, which may occur sporadically and vary seasonally. Chronic disease data were also collected, as individuals with pre-existing conditions may be more vulnerable to communicable diseases due to compromised immunity or poor health resilience, which is relevant in evaluating WASH-related risks. PEPSEP voucher adequacy - participants were also asked whether they received support through PEPSEP (Health and Nutrition Voucher Scheme for Poor, Extreme Poor, and Socially Excluded People), a

PLOS · Neglected Tropical Diseases

national initiative aimed at improving healthcare access and nutrition for vulnerable groups. This variable helped assess whether social protection mechanisms influenced health outcomes among high-risk populations..

## Statistical analyses

All data were analysed using SPSS (version 25). Descriptive statistics, including frequencies and percentages, were used to summarise the variables. Chi-square tests were initially conducted to explore bivariate associations between sociodemographic and hygiene-related variables with communicable disease status, helping to identify potential factors of interest. To further assess these associations while adjusting for confounders, a multinomial logistic regression model was employed. This model allowed us to evaluate the independent effects of multiple predictors on the likelihood of having a communicable disease. Confounders were selected based on theoretical relevance (e.g., age, gender, education, income), previous literature, and variables showing significant or near-significant associations in bivariate analyses. Before inclusion in the final model, all variables were tested for multicollinearity using variance inflation factors (VIF), and no serious multicollinearity was observed. All statistical tests were two-tailed, with a significance level set at $p < 0.05$.

## Result

Table 1 presents the sociodemographic characteristics of the 607 participants involved in this study. The majority of respondents were aged 30 years or below (40.5%), followed closely by those in the 31–40 age groups (36.7%). Females and Muslims constituted significant portions of the sample (83.7% and 87.3%, respectively). Educational attainment varied with 23.1% reporting no formal education, 51.7% completing primary education, and 16.5% finishing junior high school.

**Table 1. Socio-demographic profile of the respondents (n = 607).**

| Variable | Categories | n (%) |
|---|---|---|
| Age(years) | ≤ 30 | 246 (40.5) |
| | 31-40 | 223 (36.7) |
| | 41-50 | 76 (12.5) |
| | 51 ≥ | 62 (10.2) |
| Gender | Male | 99 (16.3) |
| | Female | 508 (83.7) |
| Religion | Muslim | 530 (87.3) |
| | Hindu | 77 (12.7) |
| Education level | No education | 140 (23.1) |
| | PSC | 314 (51.7) |
| | JSC | 100 (16.5) |
| | SSC and above | 53 (8.7) |
| Employment status | Housewife | 384 (63.3) |
| | Service holder (Gov & private, garment worker) | 63 (10.4) |
| | Labourer | 113 (18.6) |
| | Others | 47 (7.7) |
| Marital status | Married | 572 (94.2) |
| | Widowed/Separated/Single/Divorced | 35 (5.8) |
| Family income | ≤ 5000 | 64 (10.5) |
| | 5001-10000 | 303 (49.9) |
| | 10001-15000 | 193 (31.8) |
| | ≥15001 | 47 (7.7) |
| Household type | Raw (Shack, Mud) | 91 (15.0) |
| | Tin shed | 208 (34.3) |
| | Semi building | 253 (41.7) |
| | Building | 55 (9.1) |

Occupationally, the majority were housewives (63.3%), followed by labourers (18.6%), with a small percentage engaged in service roles (10.4%) and others (7.7%). Marital status was prevalent among the participants, with 94.2% being married. Half of the participants' monthly income was around 5001–10,000 taka, 31.8% earned 10001–15,000 taka, 10.5% made around 5000 taka or less, and only 7.7% earned more than 15,000 taka.

Regarding housing, the majority lived in tin shed homes (34.3%) or semi-built structures (41.7%), while only a minority resided in more modern building-type accommodations (9.1%).

Table 2 shows the hygiene and disease-related factors of the study participants. For the majority of respondents, tube well water (88.1%) met the need for drinking water. Other sources like supply water constituted the remaining sources of drinking water. The analysis also revealed the diverse sanitation practices among participants, with the majority (66.1%) utilising sanitary toilets, while 33.9% opted for water-sealing slab toilets. Notably, nearly all participants reported regular hand-washing habits, with 94.6% washing before or after food, and 95.6% practicing hand hygiene both before and after using the toilet. Waste disposal methods varied, with 46.8% disposing of household waste in dustbins, subsequently managed by municipal workers, while 53.2% of the waste was directly dumped in other ways. Regarding health factors, a substantial proportion of households (42.8%) reported at least one member with a chronic illness in the past 12 months, while 24.1% of households had at least one member affected by a communicable disease during the same period. These

**Table 2. Hygiene-related and disease factors of the study participants (n = 607).**

| Variable | Categories | n (%) |
|---|---|---|
| **Source of drinking water** | Tube well<br>Others | 535 (88.1)<br>72 (11.9) |
| **Type of toilet** | Sanitary latrine<br>Ring slab with water seal | 401 (66.1)<br>206 (33.9) |
| **Hand washing practice (before or after food)** | Yes<br>No | 574 (94.6)<br>33(5.4) |
| **Hand washing practice (before or after toilet)** | Yes<br>No | 580 (95.6)<br>27 (4.4) |
| **Garbage dump** | Household dustbin/<br>collected by municipality<br>person<br>Others | 284 (46.8)<br>323 (53.2) |
| **History of chronic disease** | Yes<br>No | 260 (42.8)<br>347 (57.2) |
| **Type of chronic disease** | Hypertension<br>Diabetes<br>Heart disease<br>Cancer<br>Stroke<br>CKD<br>Asthma | 150 (24.7)<br>146 (24.1)<br>30 (4.9)<br>1 (0.2)<br>10 (1.6)<br>15 (2.5)<br>37 (6.1) |
| **History of communicable disease** | Yes<br>No | 146 (24.1)<br>461 (75.9) |
| **Regular visits of PEPSEP volunteers** | Yes<br>No | 605 (99.7)<br>2 (0.3) |
| **Adequacy of PEPSEP voucher** | Yes<br>No | 479 (78.9)<br>128 (21.1) |

*"Supply water" refers to piped water supplied by the municipal authority, accessed directly from a tap or via connected storage. "Sanitary toilets" refer to improved sanitation facilities that safely separate human excreta from human contact. In this study, this typically included flush toilets or pit latrines with slabs and proper covers, aligned with JMP's criteria.

conditions were reported by the respondent as a representative of the household. Most (99.7%) received regular visits from PEPSEP volunteers, although opinions on the sufficiency of PEPSEP vouchers were mixed, with 78.9% considering them adequate and 21.1% considering them insufficient.

Fig 1 illustrates the distribution of specific communicable diseases among participants who reported being affected. Among those with communicable diseases (24.1% of the total sample), diarrhea was the most prevalent (12.9%), followed by typhoid (5.9%), jaundice (4%), tuberculosis (1.8%), dengue (1.3%), COVID-19 (0.8%), HIV/AIDS (0.3%), and other diseases (4.8%), respectively.

Table 3 presents the characteristics of the respondents and their correlation with communicable diseases. The results revealed several significant associations, including family income (p=0.003), history of chronic diseases (p<0.001), and source of drinking water (p=0.001). Additionally, various hygiene-related variables were significantly associated with communicable diseases, including toilet type (p=0.013), hand washing practices before and after food intake (p<0.001), hand washing practices before and after toilet use (p<0.001), and garbage dumping practices (p<0.001).

Logistic regression analysis identified several significant associations between the explanatory factors and the likelihood of developing communicable diseases (Table 4). Gender was a significant factor, with females being 3.21 times more likely than males to develop the disease (OR: 3.21, 95% CI: 1.19 to 8.66, p=0.003). The source of drinking water was also significant, with those using tube well water having higher odds ratios (OR: 2.81, 95% CI: 1.13 to 7.02, p=0.015). The toilet type showed significant associations with sanitary latrines (OR: 0.18, 95% CI: 0.07 to 0.49, p=0.002) and sanitary latrines connected to septic tanks, reducing the likelihood of disease. Unwashing hands before or after eating (OR: 23.1, 95% CI: 1.98 to 69.41, p=0.003) and not washing hands before or after using the toilet (OR: 12.31, 95% CI: 2.86 to 53.03, p<0.001) significantly increased the risk. A history of chronic disease also increased the odds of developing communicable diseases (OR, 3.73; 95% CI: 1.49 to 9.59, p=0.007).

## Discussion

This study adds to growing evidence of the association between hygiene-related factors and communicable diseases in urban Bangladeshi populations. By focusing on vulnerable households in two municipalities, the findings offer valuable insight into WASH-related disease patterns beyond rural settings where most previous work has focused. The present

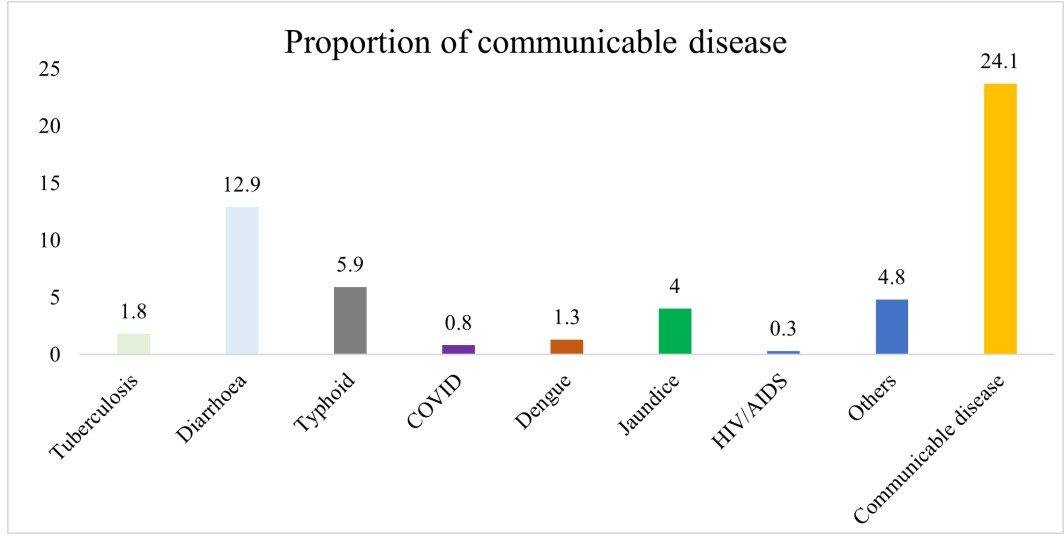

**Fig 1. Proportion of communicable diseases among the participants (N=146; 24.1% of total sample).**

**Table 3. Socioeconomic factors and their association with communicable disease (n=607).**

| Variable | Communicable disease | | Total | p-value |
|---|---|---|---|---|
| | Yes | No | | |
| **Age (In years)** | | | | 0.533 |
| ≤30 | 61 (24.8) | 185 (75.2) | 246 (40.5) | |
| 31-40 | 49 (22.0) | 174 (78.0) | 223 (36.7) | |
| 41-50 | 17 (22.4) | 59 (77.6) | 76 (12.5) | |
| 51 ≥ | 19 (30.6) | 43 (69.4) | 62 (10.2) | |
| **Gender** | | | | 0.080 |
| Male | 17 (17.2) | 82 (82.8) | 99 (16.3) | |
| Female | 129 (25.4) | 379 (74.6) | 508 (83.7) | |
| **Education level** | | | | 0.170 |
| No education | 35 (25.0) | 105 (75.0) | 140 (23.1) | |
| PSC | 65 (20.7) | 249 (79.3) | 314 (51.7) | |
| JSC | 30 (30.0) | 70 (70.0) | 100 (16.5) | |
| SSC and above | 16 (30.2) | 37 (76.3) | 53 (8.7) | |
| **Employment status** | | | | 0.796 |
| Housewife | 94 (24.5) | 290 (75.5) | 384 (63.3) | |
| Service holder | 17 (27.0) | 46 (73.0) | 63 (10.4) | |
| Laborer | 26 (23.0) | 87 (77.0) | 113 (18.6) | |
| Others | 9 (19.1) | 38 (80.9) | 47 (7.7) | |
| **Marital status** | | | | 0.164 |
| Married | 141 (24.7) | 431 (75.3) | 572 (94.2) | |
| Widowed/Separated/Single/Divorced | 5 (14.3) | 30 (85.7) | 35 (5.8) | |
| **Family income** | | | | **0.003** |
| ≤5000 | 6 (9.4) | 58 (90.6) | 64 (10.5) | |
| 5001-10000 | 68 (22.4) | 235 (77.6) | 303 (49.9) | |
| 10001-15000 | 55 (28.5) | 138 (71.5) | 193 (31.8) | |
| 15001 ≥ | 17 (36.2) | 30 (63.8) | 47 (7.7) | |
| **Household type** | | | | 0.194 |
| Raw (Shack, Mud) | 30 (33.0) | 61 (67.0) | 91 (15) | |
| Tin shed | 46 (22.1) | 162 (77.9) | 208 (34.3) | |
| Semi building | 58 (22.9) | 195 (77.1) | 253 (41.7) | |
| Building | 12 (21.8) | 43 (78.2) | 55 (9.1) | |
| **Regular visits of PEPSEP volunteers** | | | | 0.425 |
| Yes | 146 (24.1) | 459 (75.9) | 605 (99.7) | |
| No | 0 (0.0) | 2 (100.0) | 2 (0.3) | |
| **Adequacy of PEPSEP voucher** | | | | 0.148 |
| Yes | 109 (22.8) | 370 (77.2) | 479 (78.9) | |
| No | 37 (28.9) | 91 (71.1) | 128 (21.1) | |
| **History of chronic disease** | | | | **<0.001** |
| Yes | 93 (35.8) | 167 (64.2) | 260 (42.8) | |
| No | 53 (15.3) | 294 (84.7) | 347 (57.2) | |
| **Source of drinking water** | | | | **0.001** |
| Tube well | 117 (21.9) | 418 (78.1) | 535 (88.1) | |
| Others | 29 (40.3) | 43 (59.7) | 72 (11.9) | |
| **Type of toilet** | | | | **0.013** |
| Ring slab with water seal | 62 (30.1) | 144 (69.9) | 206 (33.9) | |

*(Continued)*

**Neglected Tropical Diseases**

PLOS

**Table 3.** (Continued)

| Variable | Communicable disease | | Total | p-value |
|---|---|---|---|---|
| Sanitary latrine | 84 (20.9) | 317 (79.1) | 401 (66.1) | |
| **Hand washing practice (before or after food)** | | | | **<0.001** |
| Yes | 117 (20.4) | 457 (79.6) | 574 (94.6) | |
| No | 29 (87.9) | 4 (12.1) | 33 (5.4) | |
| **Hand washing practice (before or after toilet)** | | | | **<0.001** |
| Yes | 122 (21.0) | 458 (79.0) | 580 (95.6) | |
| No | 24 (88.9) | 3 (11.1) | 27 (4.4) | |
| **Garbage dump** | | | | **<0.001** |
| Household dustbin/collected by municipality person | 48 (16.9) | 236 (83.1) | 284 (46.8) | |
| Others | 98 (30.3) | 225 (69.7) | 323 (53.2) | |

study assessed the association between hygiene-related factors and communicable diseases among the people of two municipalities in Bangladesh. To the best of our knowledge, this is one of the pioneer research studies to explore the association of hygiene-related factors with various communicable diseases in Bangladeshi urban settings. The findings of this study highlight the prevalence and correlation of communicable diseases among a sample of Bangladeshi population.

The study revealed that almost one-fourth of the participants had a communicable disease, with diarrhoea being the most common disease, followed by typhoid and jaundice.This information was self-reported by participants during face-to-face interviews, based on their disease history in the past year. These findings are consistent with previous studies conducted in Bangladesh that reported high rates of diarrhoeal diseases [33]. The study also identified several significant predictors of communicable diseases, including sex, source of drinking water, type of toilet, and hand hygiene practices. Specifically, females were more likely to develop communicable diseases than males in the current study. These results are consistent with the claims reported in existing literature [34]. A previous study reported a higher prevalence of diarrhoea among adult females than among adult males. This could be because adult females are more frequently involved in household activities, such as cooking, cleaning, and caring for children, which increases their exposure to unsafe water and poor sanitation conditions in the domestic environment [35]. These findings are consistent with results from the WASH Benefits trial in rural Bangladesh, which showed that improvements in water quality, sanitation, and hand washing significantly reduced diarrhoea and child growth faltering [36]. However, the trial also noted that single-component interventions were often insufficient on their own, highlighting the need for integrated WASH strategies, a conclusion aligned with our study's findings from urban settings.

Our research findings show that, not washing hands before or after eating or using the toilet increases the risk of communicable diseases, highlighting the importance of promoting and improving hand hygiene practices among the population [37]. Most respondents in this study practiced hand washing before and after taking food and using the toilet, but it is essential to practice hand washing regularly before and after taking food and using the toilet to prevent disease. This study indicates that those who did not maintain their hand hygiene practices had significantly higher odds of developing communicable diseases. This result is consistent with findings from research conducted in Dhaka, which revealed that the majority of their study participants washed their hands with water, but only 22.5% washed their hands effectively by maintaining the correct steps and frequency of hand washing with water and soap or hand sanitisers [38].

The findings also claimed that the use of ring slab toilets and sanitary toilets connected to septic tanks is protective against communicable diseases, which is consistent with other studies [39]. These toilets are designed to prevent disease transmission by minimising human contact with faecal matter, which can contain pathogens. According to the CDC, proper sanitation facilities (e.g., toilets and latrines) promote health because they allow people to dispose of their waste appropriately, preventing contamination of their environment and reducing risks to themselves and their

**Table 4. Association of different communicable diseases with explanatory factors identified from logistic regression.**

| Variable | | AOR | 95% CI | P-value |
|---|---|---|---|---|
| Age category | 30 or below(ref) | | | |
| | 31-40 | 2.06 | 0.94to 4.51 | 0.07 |
| | 41-50 | 2.48 | 0.98 to 5.82 | 0.112 |
| | 51 or more | 4.67 | 0.85 to 11.3 | 0.20 |
| Gender | Female | 3.21 | 1.19 to 8.66 | 0.003 |
| | Male (ref) | | | |
| Occupation | Housewife | 1.81 | 0.52 to 2.56 | 0.72 |
| | Service holder | 2.07 | 0.58 to 2.76 | 0.56 |
| | Laborer | 1.56 | 0.94 to 4.81 | 0.65 |
| | Others (ref) | | | |
| Education | Illiterate(ref) | | | |
| | PSC | 1.75 | 0.32 to 1.57 | 0.44 |
| | JSC | 2.02 | 0.42 to 2.05 | 0.25 |
| | SSC and above | 2.46 | 0.74 to 2.98 | 0.19 |
| Marital status | Married | 2.14 | 0.87 to 5.65 | 0.65 |
| | Single (ref) | | | |
| Income | 5000 or below(ref) | | | |
| | 5001-10000 | 2.94 | 0.62 to 3.28 | 0.23 |
| | 10001-15000 | 3.21 | 0.78 to 3.76 | 0.19 |
| | 15001 or more | 1.18 | 0.55 to 1.57 | 0.32 |
| Household type | Raw(ref) | | | |
| | Tin shed | 1.54 | 0.72 to 2.19 | 0.27 |
| | Semi building | 2.09 | 0.65 to 4.34 | 0.45 |
| | Building | 1.76 | 0.89 to 2.76 | 0.49 |
| Source of drinking water | Tube well | 2.81 | 1.13 to 7.02 | 0.015 |
| | Others (ref) | | | |
| Type of toilet | Sanitary latrine | 0.18 | 0.07 to 0.49 | 0.002 |
| | Ring slab with water seal (ref) | | | |
| Hand washing before or after eating | Yes (ref) | | | |
| | No | 23.1 | 1.98 to 69.41 | 0.003 |
| Hand washing before or after using toilet | Yes (Ref) | | | |
| | No | 12.31 | 2.86 to 53.03 | 0.000 |
| Household garbage dumping | Household dustbin/collected by municipals | 3.47 | 0.98 to 4.78 | 0.154 |
| | Others (ref) | | | |
| History of chronic disease | Yes | 3.73 | 1.49 to 9.59 | 0.007 |
| | No (ref) | | | |

AOR, Odds ratio (Adjusted); CI, confidence interval; ref, reference categories; P value <0.05 were found to be statistically significant.

neighbours [40]. The present study reported that household source of drinking water is associated with having communicable disease among the study participants. The present study reported that household sources of drinking water were associated with communicable diseases among study participants. According to a recent study conducted in Bangladesh, many water sources had fecal contamination with pathogens, including E. coli [13]. Also, tubewell water was found to be contaminated with Rotavirus, Adenovirus, E. Coli, Shigella, Vibrio cholerae, and other fecal pathogens [41]. Tubewells commonly found in rural areas of Bangladesh are often situated close to the latrines and ponds. Possible

mechanisms for tube well contamination with faecal pathogens include infiltration into groundwater from nearby latrines, septic tanks, and ponds [41].

This study also found that individuals with chronic illnesses were more likely to report communicable diseases. This may be due to weakened immune systems or compromised bodily functions, which can make individuals with chronic conditions more susceptible to infections. In turn, the presence of infections can exacerbate the complications of chronic illnesses, resulting in worse health outcomes. This finding supports the need for targeted preventive measures among people with preexisting health conditions to reduce their risk of communicable diseases.

Based on these findings, future WASH interventions in urban Bangladesh should prioritise safe water infrastructure, especially improving deep tube well quality control and distance from latrines or waste sources. Promotion of correct handwashing technique using soap, and upgrading sanitation facilities (e.g., from basic to improved/septic tank systems) should be central to urban public health campaigns. Community health workers and existing platforms like PEPSEP could be mobilised to deliver targeted WASH messaging, especially to households with chronic illness cases, who are at elevated risk of infection.

The strengths of this study include the use of population-based large survey data with poor municipal people who are vulnerable to developing communicable diseases and the use of rigorous statistical methods.

## Limitation

In addition to its strengths, this study had several limitations. As the survey was restricted to only two municipal areas in Dhaka and Satkhira, it did not reflect the Bangladeshi population as a whole. While socio-demographic variables such as sex, education, and income were included in the analysis, the relatively homogenous nature of the sample may limit the generalisability and depth of subgroup comparisons. Additionally, disease data were based on self-report by household representatives and were not verified through medical records, which may introduce recall bias or misclassification of disease status. As a result, estimates of disease burden may be subject to measurement error. Finally, due to the cross-sectional design, the study can identify associations but cannot establish causal relationships between hygiene-related exposures and communicable disease outcomes. Future nationwide studies using longitudinal or mixed-method designs are recommended to validate and expand these findings.

## Conclusions

In conclusion, this study highlights a significant burden of communicable diseases among urban populations in Bangladesh, with higher risk observed among females and individuals using deep tube well water. Conversely, good hand hygiene practices were associated with a lower risk of disease. These findings underscore the need for targeted interventions aimed at improving access to safe drinking water, promoting proper sanitation, and reinforcing hand hygiene behaviours, particularly among vulnerable groups. The results of this study can inform stakeholders such as policymakers, public health officials, and development agencies in designing evidence-based interventions, including WASH education campaigns, infrastructure development, and community-level health programs. Incorporating hygiene-focused strategies into national and local health plans may reduce disease burden and support long-term health outcomes. Further research across diverse regions of the country is recommended to validate and expand upon these findings, ultimately guiding a more coordinated public health response to communicable disease prevention in Bangladesh.

## Supporting information

**S1 Fig. Proportion of communicable diseases among the participants (N = 146).** This figure shows the distribution of self-reported communicable diseases among affected participants. Diarrhea was the most reported illness, followed by typhoid, jaundice, tuberculosis, dengue, COVID-19, and HIV/AIDS.
(TIF)

**S1 Table. A.** Socio-demographic characteristics of the study participants (n = 607). This table presents participants' age, gender, religion, education, occupation, marital status, income, and housing type. **B.** Hygiene-related and disease-related variables (n = 607). Details participants' water source, toilet type, handwashing practices, garbage disposal methods, chronic and communicable disease history, and use of the PEPSEP voucher program. **C.** Bivariate associations between sociodemographic and hygiene-related variables and communicable disease status (n = 607). Displays chi-square test results indicating significant and non-significant associations between selected variables and communicable disease occurrence. **D.** Multivariable logistic regression analysis identifying predictors of communicable diseases (n = 607). Presents adjusted odds ratios and confidence intervals for key variables such as gender, toilet type, water source, handwashing, and chronic illness.
(DOCX)

**S1 Questionnaire. Household questionnaire used for data collection.** The structured questionnaire was developed based on WHO's Core Questions on WASH for Household Surveys and translated into Bengali.
(PDF)

**S1 Data. The dataset used for analysis is provided in the supporting files.**
(SAV)

## Author contributions

**Conceptualization:** Mohammad Hayatun Nabi, Mosharop Hossian.

**Data curation:** Mohammad Hayatun Nabi, Fatema Hashem Rupa, Ishrat Jahan, A. B. M. Nahid Hasan, Farah Naz, Iqbal Masud.

**Formal analysis:** Mohammad Hayatun Nabi, Fatema Hashem Rupa, Mosharop Hossian.

**Funding acquisition:** Mohammad Delwer Hossain Hawlader.

**Investigation:** Fatema Hashem Rupa, Ishrat Jahan, A. B. M. Nahid Hasan, Iqbal Masud.

**Methodology:** Mohammad Hayatun Nabi, Fatema Hashem Rupa, Farah Naz, Mosharop Hossian.

**Project administration:** Mohammad Hayatun Nabi, Mohammad Delwer Hossain Hawlader.

**Resources:** Mohammad Hayatun Nabi, Fatema Hashem Rupa, Farah Naz, Mohammad Delwer Hossain Hawlader.

**Software:** Mosharop Hossian.

**Supervision:** Mohammad Hayatun Nabi.

**Validation:** A. B. M. Nahid Hasan, Iqbal Masud, Mosharop Hossian.

**Visualization:** Mosharop Hossian.

**Writing – original draft:** Mohammad Hayatun Nabi, Fatema Hashem Rupa, Ishrat Jahan, A. B. M. Nahid Hasan.

**Writing – review & editing:** Mohammad Hayatun Nabi, Fatema Hashem Rupa, Mohammad Delwer Hossain Hawlader, Mosharop Hossian.

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
