## [Decision Letter · Decision Letter 0]

Urban Hygiene and the Incidence of Communicable Diseases: A Study of Bangladeshi Municipal Areas

Dear Dr. Hashem,

Thank you for submitting your manuscript to PLOS Neglected Tropical Diseases. After careful consideration, we feel that it has merit but does not fully meet PLOS Neglected Tropical Diseases's publication criteria as it currently stands. Therefore, we invite you to submit a revised version of the manuscript that addresses the points raised during the review process.

Please submit your revised manuscript within 60 days on May 1, 2025. If you will need more time than this to complete your revisions, please reply to this message or contact the journal office at plosntds@plos.org. Please include the following items when submitting your revised manuscript:

We look forward to receiving your revised manuscript.

Kind regards,

Dawit Getachew Gebeyehu, MPH

Guest Editor

Alexandra Heaney

Section Editor

Shaden Kamhawi

co-Editor-in-Chief

Paul Brindley

co-Editor-in-Chief

**Journal Requirements:**

At this stage, the following Authors/Authors require contributions: Mohammad Hayatun Nabi, Fatema Hashem Rupa, Ishrat Jahan, A. B. M. Nahid Hasan, Farah Naz, Iqbal Masud, Mohammad Delwer Hossain Hawlader, and Mosharop Hossian. Please ensure that the full contributions of each author are acknowledged in the "Add/Edit/Remove Authors" section of our submission form.

3) Please upload the main figure as a separate Figure file in .tif or .eps format. For more information about how to convert and format your figure files please see our guidelines: 

4) Please ensure that all Figure files have corresponding citations and legends within the manuscript. Currently, Figure 01 in your submission file inventory does not have an in-text citation. If the figure is no longer to be included as part of the submission, please remove it from the file inventory.

5) In the online submission form, you indicated that "Data will be available from corresponding author on reasonable request." All PLOS journals now require all data underlying the findings described in their manuscript to be freely available to other researchers, either

1. In a public repository

2. Within the manuscript itself

3. Uploaded as supplementary information.

**Reviewers' Comments:**

**Comments to the Authors: **

**Please note that two reviews are uploaded as attachments.**

Reviewer's Responses to Questions

**Key Review Criteria Required for Acceptance?**

**Methods**

-Are the objectives of the study clearly articulated with a clear testable hypothesis stated?

-Is the study design appropriate to address the stated objectives?

-Is the population clearly described and appropriate for the hypothesis being tested?

-Is the sample size sufficient to ensure adequate power to address the hypothesis being tested?

-Were correct statistical analysis used to support conclusions?

-Are there concerns about ethical or regulatory requirements being met?

Reviewer #1: (No Response)

Reviewer #2: A description of the study sites and why they were selected would be useful.

L. 101-102: further details should be provided on the sampling procedure in order to clarify how "poor, extremely poor, marginalised and socially excluded" individuals were identified to participate in the survey and how households were located by the survey teams. The definition of these categories should also be clarified (what are income levels of "poor" and "extremely poor" households? Providing this information would help contextualise findings in the Results section).

Questionnaires should be provided as supplementary information together with the manuscript - and anonymised data.

L. 130: what is meant by household type? What about household size? who were the preferred/targeted interviewees within each household (e.g. head of household)?

L. 131: consider replacing with "WASH practices", and indicate whether access and/or practices were assessed.

L. 133-134: a recall period of 1 year is long and prone to bias; please provide a justification for this. The relevance of chronic diseases in relation to WASH should be clarified. The abbreviation "PEPSEP" needs to be defined and additional background provided.

The obtention of written consent is mentioned several times.

L. 139: "confidence" or "confidentiality"?

L. 140-146: the rationale for using both Chi-square tests and a logistic regression is unclear. The analysis should focus on answering the research questions defined in the manuscript. The logistic regression alone might be more appropriate than repeating Chi-square tests for each variable; the confounders included in the analysis should be clearly stated, and a rationale for selecting variables to be included in the model should be given (including consideration of potential collinearity between variables).

Reviewer #3: Yes

**Results**

-Does the analysis presented match the analysis plan?

-Are the results clearly and completely presented?

-Are the figures (Tables, Images) of sufficient quality for clarity?

Reviewer #1: (No Response)

Reviewer #2: It would be useful to align WASH terminology with the definitions of water and sanitation used for Sustainable Development Goal #6 (WHO/UNICEF Joint Monitoring Programmes) and/or explain how WASH solutions and terms that might be used locally correspond to the WASH ladders.

L. 161-162: "supply water" is unclear - is this referring to piped water?

L. 163-164: "sanitary toilets" is unclear

For the health-related results (L. 168-170), clarity is needed regarding the time period considered (is it the 1 year referenced in the methods?). It should also be clear whether the disease affected any member of the household or only the survey respondent.

Table 3: if kept, report the p-value as smaller than a threshold rather than zero.

Reviewer #3: Yes

**Conclusions**

-Are the conclusions supported by the data presented?

-Are the limitations of analysis clearly described?

-Do the authors discuss how these data can be helpful to advance our understanding of the topic under study?

-Is public health relevance addressed?

Reviewer #1: (No Response)

Reviewer #2: The discussion includes information that might be better placed in introduction and fails to provide clear recommendations for future WASH interventions.

Linking findings to the WASH'Benefits trial, a major intervention study which assessed WASH interventions sin Bangladesh, would be useful, too.

Limitations related to the assessment of diseases burden based on self-report should be acknowledged.

L. 266-267: this does not seem to be supported by the survey data (cross-sectional).

L. 286-288: the statement should be revised - the study design did not allow to establish causal relationships.

Reviewer #3: Yes

**Editorial and Data Presentation Modifications?**

Reviewer #1: (No Response)

Reviewer #2: The introduction can be streamlined to focus on the burden of WASH-preventable diseases such as diarrhoeal diseases and lower respiratory infections (e.g. using the Global Burden of Disease studies and national statistics). It should also include a clear description of WASH access in Bangladesh (% access to different services, differences urban/rural if relevant - where are the most important gaps).

The title only mentions hygiene but the scope of the survey covers water, sanitation and hygiene - consider adapting accordingly.

Wikipedia (ref. 7) should preferably not be included among references.

Reviewer #3: (No Response)

**Summary and General Comments**

Reviewer #1: (No Response)

Reviewer #2: The manuscript reports data from a cross-sectional survey of >600 households assessing disease burden and WASH access in two communities in Bangladesh. The study could potentially help identify risk factors for infectious diseases in vulnerable populations. However, I have concerns about (i) the disease burden assessment based on self-report, with a recall period of 1 year, which is likely unreliable; (ii) the statistical analyses, using both (repeated) Chi-square tests and logistic regression, with a lack of clarity regarding the tested hypotheses; (iii) the discussion, which fails to provide clear recommendations based on the survey findings.

Reviewer #3: (No Response)

PLOS authors have the option to publish the peer review history of their article (what does this mean? ). If published, this will include your full peer review and any attached files.

**Do you want your identity to be public for this peer review?** For information about this choice, including consent withdrawal, please see our Privacy Policy .

Reviewer #1: No

Reviewer #2: No

Reviewer #3: No

**Figure resubmission:**

**Reproducibility:**



---

## [Decision Letter · Decision Letter 1]

Dear Fatema,

We are pleased to inform you that your manuscript 'Water, Sanitation, and Hygiene (WASH) Factors and the Incidence of Communicable Diseases in Urban Bangladesh: Evidence from Municipal Areas.' has been provisionally accepted for publication in PLOS Neglected Tropical Diseases.

Best regards,

Dawit Getachew Gebeyehu, MPH

Guest Editor

Alexandra Heaney

Section Editor

Shaden Kamhawi

co-Editor-in-Chief

Paul Brindley

co-Editor-in-Chief

Reviewer's Responses to Questions

**Key Review Criteria Required for Acceptance?**

**Methods**

-Are the objectives of the study clearly articulated with a clear testable hypothesis stated?

-Is the study design appropriate to address the stated objectives?

-Is the population clearly described and appropriate for the hypothesis being tested?

-Is the sample size sufficient to ensure adequate power to address the hypothesis being tested?

-Were correct statistical analysis used to support conclusions?

-Are there concerns about ethical or regulatory requirements being met?

Reviewer #1: (No Response)

Reviewer #2: Comments were well addressed.

**Results**

-Does the analysis presented match the analysis plan?

-Are the results clearly and completely presented?

-Are the figures (Tables, Images) of sufficient quality for clarity?

Reviewer #1: (No Response)

Reviewer #2: Comments were well addressed.

**Conclusions**

-Are the conclusions supported by the data presented?

-Are the limitations of analysis clearly described?

-Do the authors discuss how these data can be helpful to advance our understanding of the topic under study?

-Is public health relevance addressed?

Reviewer #1: (No Response)

Reviewer #2: (No Response)

**Editorial and Data Presentation Modifications?**

Reviewer #1: (No Response)

Reviewer #2: Comments were well addressed.

**Summary and General Comments**

Reviewer #1: (No Response)

Reviewer #2: The manuscript has improved and comments were addressed satisfactorily.

PLOS authors have the option to publish the peer review history of their article (what does this mean? ). If published, this will include your full peer review and any attached files.

**Do you want your identity to be public for this peer review?** For information about this choice, including consent withdrawal, please see our Privacy Policy .

Reviewer #1: **Yes: ** Firdausi Qadri

Reviewer #2: No

---

## [Editor Report · Acceptance letter]

Dear Dr Rupa,

We are delighted to inform you that your manuscript, "Water, Sanitation, and Hygiene (WASH) Factors and the Incidence of Communicable Diseases in Urban Bangladesh: Evidence from Municipal Areas.," has been formally accepted for publication in PLOS Neglected Tropical Diseases.

Best regards,

Shaden Kamhawi

co-Editor-in-Chief

Paul Brindley

co-Editor-in-Chief
